# MeshMask: Physics-Based Simulations with Masked Graph Neural Networks

**Paul Garnier**[*]
Mines Paris - PSL University
Centre for Material Forming (CEMEF)
CNRS
`paul.garnier@minesparis.psl.eu`

**Vincent Lannelongue**
Mines Paris - PSL University
Centre for Material Forming (CEMEF)
CNRS
`vincent.lannelongue@minesparis.psl.eu`

**Jonathan Viquerat**
Mines Paris - PSL University
Centre for Material Forming (CEMEF)
CNRS
`jonathan.viquerat@minesparis.psl.eu`

**Elie Hachem**
Mines Paris - PSL University
Centre for Material Forming (CEMEF)
CNRS
`elie.hachem@minesparis.psl.eu`

## Abstract

We introduce a novel masked pre-training technique for graph neural networks (GNNs) applied to computational fluid dynamics (CFD) problems. By randomly masking up to 40% of input mesh nodes during pre-training, we force the model to learn robust representations of complex fluid dynamics. We pair this masking strategy with an asymmetric encoder-decoder architecture and gated multi-layer perceptrons to further enhance performance. The proposed method achieves state-of-the-art results on seven CFD datasets, including a new challenging dataset of 3D intracranial aneurysm simulations with over 250,000 nodes per mesh. Moreover, it significantly improves model performance and training efficiency across such diverse range of fluid simulation tasks. We demonstrate improvements of up to 60% in long-term prediction accuracy compared to previous best models, while maintaining similar computational costs. Notably, our approach enables effective pre-training on multiple datasets simultaneously, significantly reducing the time and data required to achieve high performance on new tasks. Through extensive ablation studies, we provide insights into the optimal masking ratio, architectural choices, and training strategies.

## 1 Introduction

The quest to efficiently supplement traditional computational fluid dynamics (CFD) Hachem et al. (2010) with machine learning techniques was marked by multiple advances in the past decade, progressing from convolutional neural network approaches Tompson et al. (2016); Thuerey et al. (2018); Chen et al. (2019); Chu & Thuerey (2017) and physics informed neural networks Raissi et al. (2019) to advanced graph neural networks (GNN) methods. Indeed, graph neural networks, and more specifically the message-passing approach Battaglia et al. (2018), have emerged as a natural candidates for the processing of mesh-based data, leading to a seamless coupling with CFD Pfaff et al. (2021).

This architecture has brought spectacular results for the auto-regressive inference of non-stationary physical phenomena Sanchez-Gonzalez et al. (2020); Lam et al. (2023). The recent introduction of attention layers and transformer architectures in GNNs has brought even more improvements, coupled with more complex training procedure such as diffusion models Price et al. (2024), although with limited mesh sizes.

---

[*]Corresponding author

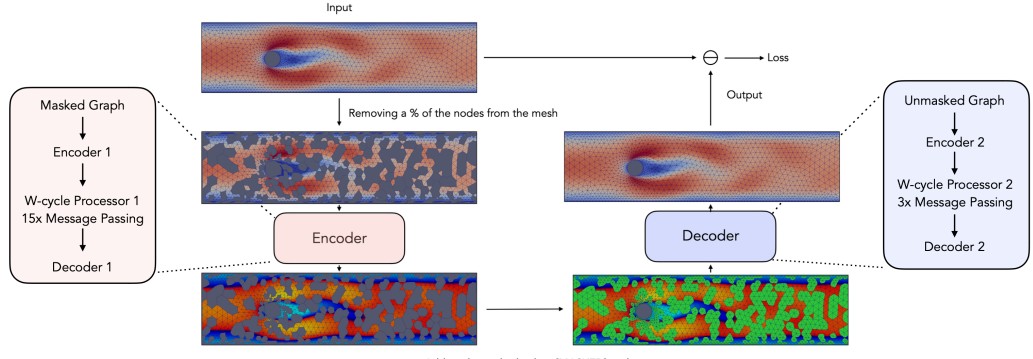

Figure 1: The proposed Masked Graph Neural Network architecture. During pre-training, we remove a portion of the edges and nodes of the mesh (*e.g.* 50%) and use this new graph as input for the Encoder. Masked nodes are then replaced by a `[MASKED]` token before the mesh is passed to the Decoder. After pretraining, only the Encoder is finetuned on a specific dataset, without any nodes being removed.

The refinement of the architectures and training techniques to overcome this limitation is still an active research field, with multi-grid paradigms John (2002) showing promising results Lino et al. (2021); Fortunato et al. (2022); Yang et al. (2022); Taghibakhshi et al. (2023); Garnier et al. (2024). Yet, training times remain a major concern, and the required amount of training data can still represent a prohibitive pre-processing cost.

Recently, pre-training techniques such as the Cloze Task Taylor (1953) have been introduced in the domain of natural language processing (NLP) and computer vision that improve performance while significantly reducing training times. Most of the contributions rely on masked auto-encoders, an asymmetric architecture in which an encoder is provided with a random subset of the input, a proportion of it being masked (usually around 70% of the data is randomly masked for computer vision and 15% for NLP). The masked tokens are re-introduced on the latent representation, after what a lightweight decoder reconstructs the full output. This pre-training step is usually followed by a fine-tuning training on specific tasks. This approach has led to significant improvements in computer vision He et al. (2021) and NLP Devlin et al. (2019); Radford et al. (2018).

Such methods have already been transferred in the context of GNNs. In Hu et al. (2019), the authors consider graph classification datasets, and develop a pre-training approach mixing both local and global representations. The authors in Hu et al. (2020) develop a generative framework for the reconstruction of the inputs, leading to significant improvements against state of the art supervised results. In Tan et al. (2022), the authors propose the masked graph auto-encoder (MGAE) paradigm that makes use of a tailored cross-correlation decoder structure. The results show the superiority of this approach for link prediction and node classification tasks. We also underline the contribution of Zhou & Farimani (2024), who successfully apply computer vision approaches to the resolution of one- and two-dimensional PDE problems, including the Burgers equation or the Kuramoto-Sivashinsky system.

In the present contribution, we propose an architecture variation with a new masking pre-training technique for CFD datasets. We also introduce a new dataset much more complex than previous ones, both in terms of physics and mesh size (up to 250,000 nodes and 3 millions edges). We conduct a comprehensive ablation study on model architecture, Encoder and Decoder asymmetry and on key parameters of our pre-training method. Finally, we train our models on 7 different datasets, all different in terms of physics, mesh size and trajectory length.

Our finding suggests that our pre-training technique gives large improvements in term of accuracy (from 15 to 40%), and more importantly, offers a very convenient method to pre-train a model on different datasets before being finetuned, making it much easier to train on large datasets.

The paper is organized as follows: the AutoEncoder architecture and masking strategy are presented in subsection 2.1 and 2.2. The overall structure of each model is defined in 2.3. Training methodology

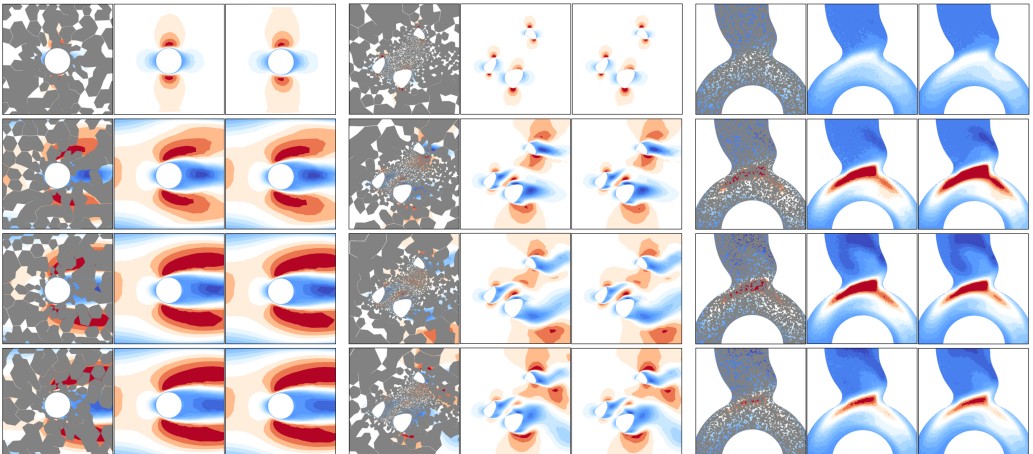

Figure 2: Examples from test meshes in 3 different datasets: CYLINDER, BEZIER and 2D-ANEURYSM. We show **(left)** the masked mesh with hidden nodes in gray, **(middle)** the prediction from the AutoEncoder architecture, and **(right)** the ground truth prediction. We replace the masked nodes by the ground truth ones.

and datasets are presented in section 3. We then perform a full ablation study in subsection 4.1 and present the rest of the obtained results in section 4. Finally, future perspectives are given. The code used in this paper will be released after publication.

## 2 THEORETICAL FRAMEWORK

We consider a mesh as an undirected graph $G = (V, E)$. $V = \{\mathbf{v}_i\}_{i=1:N^v}$ is the set of nodes, where each $\mathbf{v}_i$ represents the attributes of node $i$. $E = \{(\mathbf{e}_k, r_k, s_k)\}_{k=1:N^e}$ is the set of edges, where each $\mathbf{e}_k$ represents the attributes of edge $k$, $r_k$ is the index of the receiver node, and $s_k$ is the index of the sender node.

Each edge feature is made of the relative displacement vector in mesh space $\mathbf{u}_{ij} = \mathbf{u}_i - \mathbf{u}_j$ and its norm $\|\mathbf{u}_{ij}\|$. Each node feature $\mathbf{v}_i$ (such as the pressure, velocity) also receives a one-hot vector indicating the node type (such as inflow or outflow for boundary conditions, obstacles to denote where shapes are inside the domain, etc) and global information (viscosity, gravity) creating $\mathbf{x}_i$. In the case of the ANEURSYM dataset, we also add the nodes positions and the acceleration as inputs. Finally, given the complexity of the blood inflow (see subsection 3.1), we also add the next step inflow velocity as inputs.

### 2.1 ENCODER-DECODER ARCHITECTURE

The proposed Encoder-Decoder architecture simply stacks 2 GNNs one after the other in an AutoEncoder fashion. Each of these two global networks is a multi-grid GNN built in an Encode-Process-Decode fashion, as shown in Figure 1 (see subsection 2.3).

The aim is to make the next-step prediction (or the reconstruction) of a mesh given a partially visible input. We follow the strategy of He et al. (2021) by having an asymetric architecture, meaning our Encoder is much larger than our Decoder. This does not increase the training budget since most of it is spent during pre-training, where the Encoder inputs are mostly masked.

In practice, we start by pre-training the Encoder and the Decoder on a next-step prediction or reconstruction task with masked input (as detailed in 2.2). We then finetune the Encoder only on a next-step prediction task. The Encoder architecture does not change between pretraining and fine-tuning.

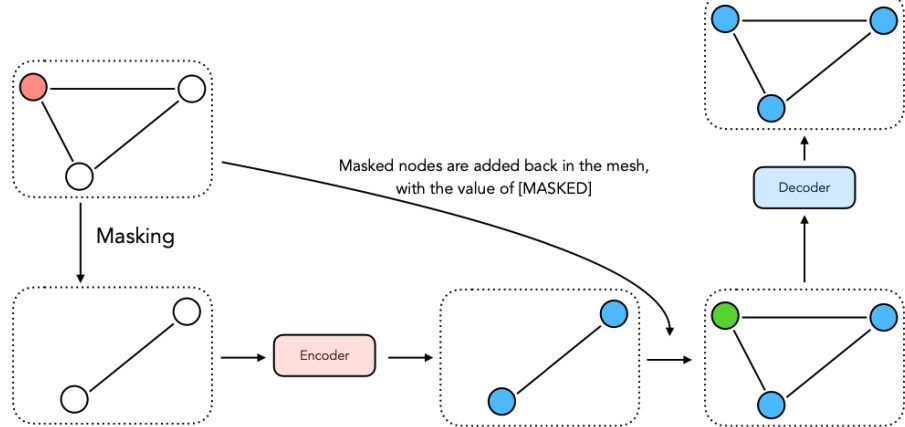

Figure 3: The overall AutoEncoder architecture.

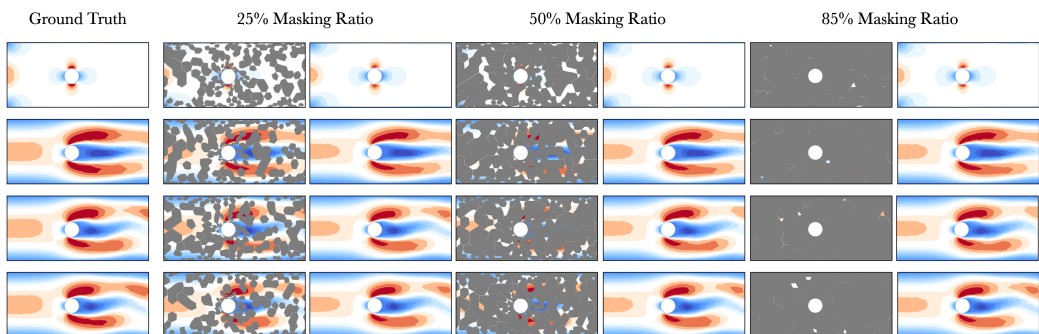

Figure 4: Predictions on the CYLINDER with different Masking Ratio. Even with a high ratio (85%), the model is able to generalize well.

## 2.2 MASKING

At each training step, we randomly sample a fraction of the existing nodes in a given mesh. We then proceed to completely remove them from the mesh, including all edges that are connected to at least one masked node. To ensure that the model could process long range interactions even with high masking ratios and disconnected sub-graphs, we considered computing the $K$-hop of each remaining node in the original graph and adding those new longer edges if both nodes were not masked, for different $K$ values.

We refer to this simply as "masking". What we end up with as inputs is now a much smaller mesh (up to 95% smaller in our experiments), thus being much faster to process. We display in Figure 4 examples of different masking ratio.

After being passed to the Encoder, we reconstruct the initial mesh by:

- Replacing hidden nodes by a shared learnable [MASKED] token (following Devlin et al. (2019))
- Replacing other nodes by their predicted values from the Encoder
- Rebuilding hidden edges with geometric data

**Theoretical justification**    This architecture is inspired by three factors: the inherent locality of finite element solvers, the information propagation boundaries of message passing, and the scale of error reduction in GNNs. First, since finite element solvers approximate solutions to equations such as the Navier-Stokes equation on neighboring elements, masking an element naturally makes it a much

more difficult task. We believe that by doing so, we force our model to find other relevant elements to make predictions rather than relying solely on direct neighbors. Second, since a message passing architecture is bounded by the number of steps multiplied by the length of edges, masking a large part of the mesh drastically reduces both the available information and how far it can propagate. We thus prevent the model from simply extrapolating from neighboring physics. Third, since GNNs can only reduce errors locally, like Gauss-Seidel smoothers, we are forcing them to retrieve information from more distant parts of the mesh than they usually do.

Finally, thanks to the unstructured nature of a mesh, uniform random masking allows the model to focus more on fine-grained area of the mesh. The overall process is detailed in Figure 3.

## 2.3 Overall architecture

Both our Encoder and Decoder follow an Encode-Process-Decode architecture, following the work from Battaglia et al. (2018). We encode nodes and edges features with 2 simple Multi Layer Perceptron (MLP) into latent vectors of size $p$: $\mathbf{e}_k = \text{MLP}(\mathbf{u}_{ij}, \|\mathbf{u}_{ij}\|) \quad \forall k \in E$, $\mathbf{v}_r = \text{MLP}(\mathbf{x}_r) \quad \forall r \in V$

where the MLP is made of 2 hidden layers of size $p$, ReLU activation and Layer Normalization.

The proposed process block is made of $m$ message passing blocks, each being either a Graph Net block from Battaglia et al. (2018) or UpScale and DownScale blocks from Garnier et al. (2024) to create Multigrid W-cycle models. If not specified, our models are W-cycle with 15 message passing, and a latent size of 128 neurons.

Usually, the processing is done by an MLP similar to the ones seen in the Encoder. Here, we improve on that architecture by replacing them with Gated MLP (see subsection 2.4). We also considered using Graph Attention Networks (GAT) as introduced by Veličković et al. (2018) for processing, since self-attentional layers have proven to be effective for such applications Garnier et al. (2024). The message passing process is detailed in Figure 5.

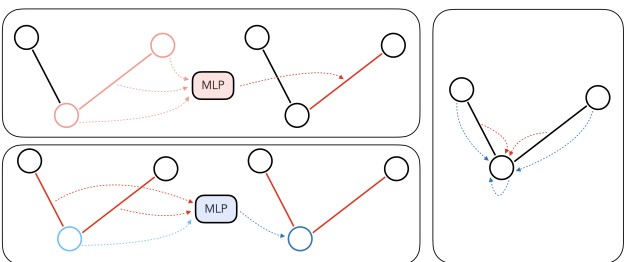

Figure 5: **(top)** Each edge is updated from its own feature, and the one from the nodes connected to it. **(bottom)** Each node is updated from its own features, and from the edges its connected to. **(right)** Information flow from a node perspective after one step of message passing.

Finally, we add a Decoder to transform the latent $\mathbf{v}$ into the output features: $\mathbf{y}_r = \text{MLP}(\mathbf{v}_r) \quad \forall r \in V$

## 2.4 Gated MLP

We replace the MLPs in the Processor by Gated MLP (see Dauphin et al. (2017)). The Gated MLP takes an input of size $p$, creates two branches where linear layers are applied with an expansion factor of size $e$ and apply the GeLU non-linearity Hendrycks & Gimpel (2023) on one of those 2 branches. We then merge the two branches with element-wise multiplication before applying a final linear layer with an output of size $p$.

While this architecture leads to a much higher number of parameters, training and inference time are similar as to their full MLP counterpart. We find this approach similar to Shazeer (2020) and that it leads to improvements in the predictions.

## 3 Training

### 3.1 Datasets

We evaluated our models on different use cases. We detailed below the different datasets (see Table 3.1 and Figure 6), parameters used and the simulation time step $\Delta t$. Each training set contains 100

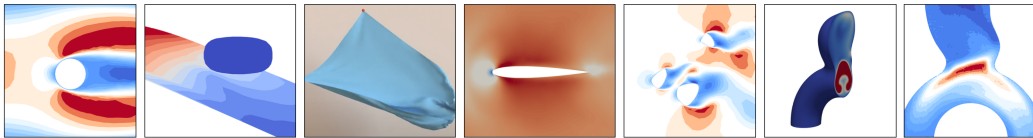

Figure 6: Sample of our datasets, in the same order as in Table 3.1.

trajectories, and testing set 20 trajectories. Datasets from the COMSOL solver are originally from Pfaff et al. (2021). The MULTIPLE BEZIER dataset is from Garnier et al. (2024). The 3D ANEURYSM dataset is from Goetz et al. (2024b). Given the considerable leap in complexity (both in terms of mesh size, inputs and physics), a more detailed presentation is given below in 3.1.1.

| Dataset | Solver | # nodes | Dimension | # traj | # steps | $\Delta t$ s |
|---------|--------|---------|-----------|--------|---------|--------------|
| CYLINDER | COMSOL | 2k | 2D Fixed Mesh | 100 | 600 | 0.01 |
| PLATE | COMSOL | 1k | 3D Fixed Mesh | 100 | 400 | - |
| FLAGSIMPLE | ArcSim | 2k | 2D Fixed Mesh | 100 | 400 | 0.02 |
| AIRFOIL | SU2 | 5k | 2D Fixed Mesh | 100 | 600 | 0.008 |
| BEZIER | Cimlib | 30k | 2D Fixed Mesh | 100 | 6000 | 0.1 |
| 3D-ANEURYSM | Cimlib | 250k | 3D Fixed Mesh | 100 | 80 | 0.01 |
| 2D-ANEURYSM | Cimlib | 10k | 2D Fixed Mesh | 100 | 80 | 0.01 |

### 3.1.1 3D ANEURYSMS

Goetz et al. (2024b) developed 101 semi-idealized geometries derived from patient-specific intracranial aneurysms, segmented from medical imaging data. They conducted CFD simulations of blood flow within these vessels, numerically solving the transient incompressible Navier-Stokes equation over a complete cardiac cycle. For a more detailed explanation of the methodology, refer to Goetz et al. (2023; 2024a).

Training on this dataset (Figure 7) represents a significant advancement and introduces unprecedented challenges. The transition to three-dimensional fluid dynamics substantially increases the complexity of observed flow patterns and expands mesh sizes to over 250,000 nodes and 3 million edges, far surpassing previous studies in this field, which typically didn't exceed 40,000 nodes. Moreover, the inflow conditions follow a pulsatile cardiac cycle, introducing time-dependent variations that require accurate modeling. This pulsatile nature demands a more profound understanding of fluid dynamics to simulate the rhythmic fluctuations in blood flow and pressure.

From the 3D-ANEURYSM dataset, we also derived the 2D-ANEURYSM dataset by re-meshing slices extracted from the 3D simulations. This dataset allowed us to iterate more quickly while retaining the challenges of varying geometry, pulsatile inflow, and complex flow patterns.

### 3.2 PARAMETERS

**Network Architecture** All of the MLPs (except the Gated MLPs) are made of 2 hidden layers of size 128 with ReLU activation functions. Outputs are normalized with a LayerNorm. The Gated MLPs are using a hidden dimension $p$ of size 128 and an expansion factor $e = 3$.

In the case of MultiGrid model, DownScale blocks use a ratio of 0.5. We follow the state-of-the-art and all our models are W-cycle with 15 message passing steps.

**Training** We trained our models using an $L_2$ loss, with a batch size of $2^1$. During pretraining, the loss is only computed on masked nodes, similar to Devlin et al. (2019); He et al. (2021).

We start by pre-training our Encoder and Decoder for 500k training steps, using an exponential learning rate decay from $10^{-4}$ to $10^{-6}$ over the last 250k steps. We then finetune the Encoder for

---

[1]In the case of the 3D ANEURYSM, we use a batch size of 1 with sub-mesh partitioning (see Appendix)

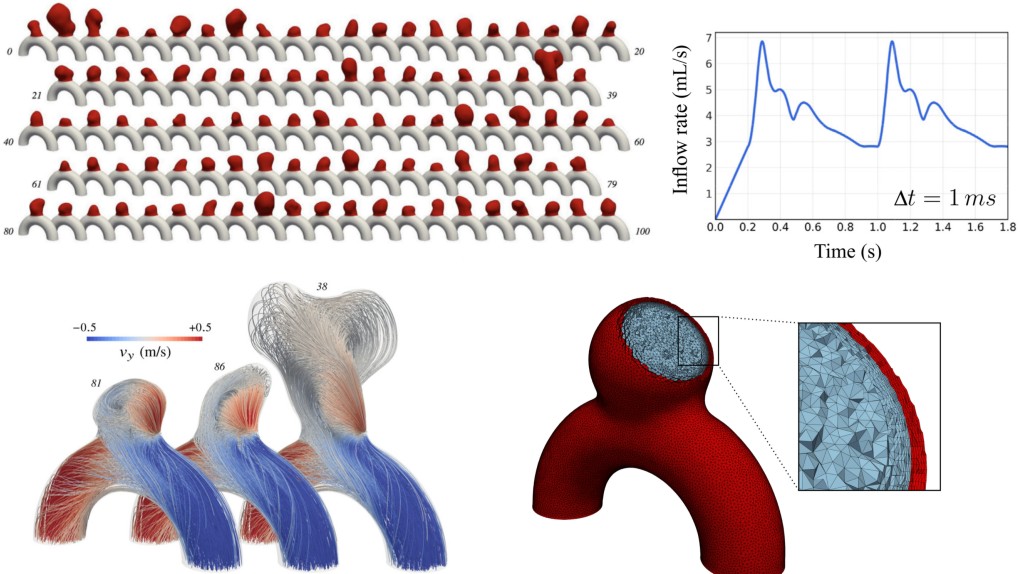

Figure 7: **(top-left)** Overview of the 101 aneurysms. **(top-right)** Velocity inflow profile imposed at the inlet boundary as a parabolic flow. **(bottom-left)** Presentation of the $v_y$ flow in 3 aneurysms. **(bottom-right)** Example of a mesh.

another 500k training steps, with the same strategy for the learning rate. During the pre-training, if not specified, we use a node masking ratio of $40\%$ (roughly equivalent to masking 60 to 70% of the mesh information depending on its geometry).

All models are trained using an Adam optimizer Kingma & Ba (2017). The baseline models are trained for a million training steps to evenly compare to our models. Finally, following the same strategy as Sanchez-Gonzalez et al. (2020); Pfaff et al. (2021); Garnier et al. (2024), we introduce noise to our inputs. More specifically, we add random noise $\mathcal{N}(0, \sigma)$ to the dynamical variables.

## 4  RESULTS

### 4.1  ABLATION STUDY

We trained our models on 7 very different datasets (results can be seen here) and compared it to baseline models as well as state-of-the-art model. We also pretrained our models on multiple datasets at the same time, before evaluating them in a 0-shot fashion and with finetuning. The main finding is that masking a large portion of the input mesh during pretraining leads to a large improvement for long-term autoregressive prediction (see table 2). We are also able to pretrain models on different datasets at the same time, largely improving the time needed to fully train a model. Finally, our models decreases inference time by a large margin in comparison to classical finite element solver (see Table A.1.1). Even when taking the training time into account, it becomes more interesting to train a GNN and use it for inference after roughly 250 simulations of Cylinder, and only 20 3D Aneurysms.[2]

**Masking Ratio**   Figure 8 shows how the masking ratio impacts the long-term autoregressive prediction. We find that the best masking ratio lies between 25% and 40%. This represents masking between 40% and 70% of the overall mesh (nodes and edges). While this is much higher than the usual 15% from Devlin et al. (2019), this is similar to the results obtained by He et al. (2021). We find that adding $K$-hop edges did not have a significant impact for masking values up to 20%, but improved performance for larger values with $K = 2$ and $K = 3$.

---

[2]For the 3D Aneurysm, it takes 12 hours to simulate a full trajectory while only 80 seconds with a GNN that took 10 days of training.

| Task | 1-step | all-rollout |
|---|---|---|
| Reconstruction | 3.49 | 86 |
| Next-step prediction | 2.49 | 52 |

(a) **Pre-training task**. Training for next-step prediction leads to much better results.

| Layer | Depth | 1-step | all-rollout |
|---|---|---|---|
| MLP | 5 | 6.23 | 185 |
| MLP | 15 | 3.13 | 72 |
| Gated MLP | 5 | 3.6 | 71 |
| Gated MLP | 15 | 2.1 | 58 |
| GAT | 15 | 2.2 | 58.2 |

(b) **Processing layers**. Gated MLP performs better.

| Masking | all-rollout | FLOPS |
|---|---|---|
| encoder w/ `[MASKED]` | 54.3 | 2.8× |
| encoder w/out `[MASKED]` | 52 | 1× |

(c) **Encoder Mask token**. No mask token makes it faster.

| Decoder Depth | 1-step | all-rollout |
|---|---|---|
| 1 | - | 49 |
| 3 | - | 46 |
| 5 | - | 45 |

(d) **Decoder Depth**. Increasing the number of message passing steps only slightly increases performances.

Table 1: **Ablation study** on the CYLINDER Dataset. We tracked one-step RMSE and the RMSE averaged over the entire trajectory. We highlight the settings chosen by our model in the following experiments. All results are $\times 10^{-3}$.

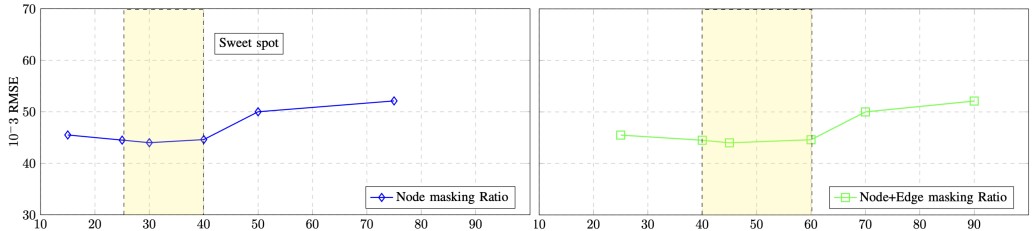

Figure 8: A masking ratio between 25% and 40% leads to the best finetuning results on the CYLIN-DERFLOW dataset.

**Impact of Masking Pretraining**   Table 2 shows the impact of masking pretraining on a simple MeshGraphNet architecture. Overall, we find that keeping the same model but pretraining it for 500k steps with a masking ratio of 40% leads to a 25% improvement consistently. This shows that our novel method allows the model to learn the inherent physics with more depth. The improvement in performances without any change in the underlying architecture nor extra training time makes it a very robust technique to systematically improve the model's performances.

**Gated MLP**   We find that using a Gated MLP instead of a regular MLP yields much better results. We find similar results for a model 3 times smaller using Gated MLP versus a model using MLP. Using attention-based processing layers here did not prove to enhance the performance compared to the Gated MLP.

At scale with 15 message passing steps, we still find improvements over the usual MeshGraphNet architecture from Pfaff et al. (2021) (see table 1b)

**Auto-Encoder Architecture**   The proposed Decoder architecture can be defined independently from the Encoder. We study this in table 1d with different amount of message passing steps. Similarly to He et al. (2021), we find that a Decoder with only one message passing step already yields very strong

performance. Overall, we select a Decoder with 3 message passing steps since the performances are on-par with bigger Decoder, but at a much lighter cost of training time.

## 4.2 COMPARISON

Our experiments demonstrate significant improvements in model performance across various architectures when employing the proposed novel pre-training masking techniques. Regardless of the underlying model, we observe performance gains of up to 25% compared to baseline models without masking (Table 2). The most striking results come from our multigrid architecture combined with masking pre-training, which establishes a new state-of-the-art in the field. This configuration outperforms the previous best model by up to 60%, marking a substantial leap forward in predictive accuracy for fluid dynamics simulations.

Notably, these improvements are achieved with the same number of training updates as baseline models, underscoring the efficiency of our masking techniques in helping the model grasp the intricate physics of fluid dynamics. This suggests that our approach not only enhances performance but also accelerates the learning process, reducing computational costs for future large-scale simulations. A comprehensive overview of our results is presented in Table 2, clearly illustrating the performance gains across different model architectures and evaluation metrics.

Interestingly, we observe minimal or no improvements in 1-step RMSE across our experiments. This observation points to two key insights: firstly, we may be approaching a performance plateau in terms of pure 1-step loss prediction. Secondly, and perhaps more significantly, our models demonstrate an improved ability to mitigate long-term error propagation, a crucial factor in extending the predictive horizon of fluid dynamics simulations.

Table 2: All numbers are $\times 10^{-3}$. DATASET-1 means one-step RMSE, and DATASET-All means all-rollout RMSE. MGN results are reproduced according to Pfaff et al. (2021), BSMS-GNN according to Cao et al. (2023), MultiScaleGNN according to Lino et al. (2021) and Multigrid results are reproduced according to Garnier et al. (2024).

| MODEL | CYLINDER 1-RMSE ↓ | SIMPLEFLAG 1-RMSE ↓ | PLATE 1-RMSE ↓ | AIRFOIL 1-RMSE ↓ | BEZIER 1-RMSE ↓ | 2D-ANEURYSM 1-RMSE ↓ | 3D-ANEURYSM 1-RMSE ↓ |
|---|---|---|---|---|---|---|---|
| MGN | 3.3 | 1.15 | **0.07** | 329 | 27.7 | 794 | 1795 |
| Multigrid | 2.8 | **1.02** | 0.17 | 302 | 24.1 | 638 | 749 |
| MultiScaleGNN | 2.7 | - | 0.10 | **300** | 24.8 | - | 823 |
| BSMS-GNN | 2.83 | - | 0.15 | 314 | 23.2 | **632** | **719** |
| MGN w/ masking | **2.5** | - | 0.09 | - | 26.5 | 718 | - |
| Ours | **2.5** | 1.04 | 0.11 | 310 | **22.7** | 645 | 725 |
| | ALL-RMSE ↓ | ALL-RMSE ↓ | ALL-RMSE ↓ | ALL-RMSE ↓ | ALL-RMSE ↓ | ALL-RMSE ↓ | ALL-RMSE ↓ |
| MGN | 71.4 | 146 | 16.9 | 11398 | 335 | 7513 | 13747 |
| Multigrid | 56.9 | 121 | 8.1 | 9871 | 275 | 7009 | 11327 |
| MultiScaleGNN | 61.2 | - | 15.7 | 10272 | 289 | - | 11612 |
| BSMS-GNN | 56.9 | - | 16.6 | 9418 | 262 | 6983 | 10993 |
| MGN w/ masking | 46.5 | - | 12.2 | - | 281 | 7112 | - |
| Ours | **29** | **98** | **4.5** | **8794** | **212** | **6489** | **8772** |

Table 3: Every model is trained for a total of 1M steps. vRAM and Inference Time are computed on the CYLINDER dataset. The large increase of parameters in our method is mostly due to the gatedMLP and the expansion factor $e = 3$.

| MODEL | # TRAINING STEPS | # PARAMETERS | vRAM (IN GB) | INFERENCE TIME (MS/STEP) |
|---|---|---|---|---|
| MGN | 1M | 2.8M | 7 | 49.3 |
| Multigrid | 1M | 3.5M | 11 | 55.2 |
| MultiScaleGNN | 1M | 2.6M | 8 | 53.7 |
| BSMS-GNN | 1M | 2.1M | 7 | 52.8 |
| MGN w/ masking | 500k + 500k | 2.8M | 7 | 49.3 |
| Ours w/ GatedMLP | 500k + 500k | 9.2M | 16 | 59.1 |

## 4.3 TRANSFER LEARNING EXPERIMENTS

**Pretraining on multiple datasets**   We evaluate transfer learning on two datasets: CYLINDER and BEZIER. Both datasets share the same inputs and outputs (see subsubsection A.1.1), and are generated from the simulation of an incompressible flow past one or multiple rigid bodies in a fixed Reynolds number range, thus sharing physical similarities. We conducted three experiments:

- masked pretraining on CYLINDER → finetuned on BEZIER
- masked pretraining on BEZIER → finetuned on CYLINDER
- masked pretraining on both CYLINDER and BEZIER before being duplicated and → finetuned on both datasets

We compare those experiments to both our best model without pretraining and our best model with pretraining. Results are available in Table 4.

We find that pretraining on a similar (although with different physical parameters such as the viscosity or the inlet velocity) dataset yields good performances similar to the models without any pretraining (+7.5% on CYLINDER, +14.5% on BEZIER).

Our pretraining methods allows for efficient pretraining on multiple datasets at the same time before re-using this model for a finetuning phase. A model pretrained on both CYLINDER and BEZIER improves its results respectively by 22.7% and 15.6%, while being 33% faster to train those 2 models. This highlights two important points:

1. This shows that our pretraining method not only allows to train multiple models faster, but also that it can lead to the pretraining of a larger model on multiple datasets of different physics use cases before being finetuned.
2. We find that pretraining, even on multiple datasets at the same time, always leads to better performances in terms of all-rollout RMSE (-18% on average) than no pretraining at all.

Table 4: All numbers are $\times 10^{-3}$. Each row presents on which dataset the model was pretrained. Each column presents on which dataset the model was finetuned and then evaluated.

| PRETRAINING DATASET | CYLINDER | | BEZIER | |
| --- | --- | --- | --- | --- |
| | ALL-RMSE ↓ | % DIFFERENCE W/ BASELINE | ALL-RMSE ↓ | % DIFFERENCE W/ BASELINE |
| NO PRETRAINING | 56.9 | - | 275 | - |
| CYLINDER | 29 | -49.0% | 315 | +14.5% |
| BEZIER | 61.2 | +7.5% | 212 | -22.9% |
| CYLINDER+BEZIER | 44 | -22.7% | 229.6 | -15.6% |

**Out-of-distribution meshes**   We evaluated the performance of our model on out-of-distribution meshes by training a model on graphs with a given refinement and applying this model directly to a much finer graph (from 10k to 250k nodes in our experiments). We find that even with this large gap in refinement, our model is able to generalize well and make accurate predictions as it performs only 75% worse than a model fully trained on fine meshes, which remains on par with results from MGN for example. These results also mean that we do not necessarily have to produce the entirety of the training on large and costly meshes, but that a fraction might be sufficient.

## 5   CONCLUSION

In this study, we show on multiple datasets and through transfer learning that similar to NLP and Computer Vision, our simple masking pretraining technique demonstrated significant improvements in model performance and training efficiency.

Our approach achieved state-of-the-art results on the seven presented CFD datasets, including a new challenging dataset of 3D intracranial aneurysm simulations. Importantly, our method allows for efficient pre-training across multiple datasets simultaneously, greatly reducing both the time and data needed to attain high performance on new tasks.

**Acknowledgements** The authors acknowledge the financial support from ERC grant no 2021-CoG-101045042, CURE. Views and opinions expressed are however those of the author(s) only and do not necessarily reflect those of the European Union or the European Research Council. Neither the European Union nor the granting authority can be held responsible for them.

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

## A APPENDIX

### A.1 DATASETS

#### A.1.1 DETAILS

We give details below about the inputs and outputs used for each dataset (see Table A.1.1). CYLINDER, PLATE were generated with COMSOL multiphysics® (2020) and were introduced by Pfaff et al. (2021). FLAGSIMPLE was generated with ArcSim Narain et al. (2012) and was introduced by Pfaff et al. (2021). AIRFOIL was generated with SU2 Economon et al. (2016) and was introduced by Pfaff et al. (2021). BEZIER was generated with CimLib Digonnet et al. (2007) ans was was introduced by Garnier et al. (2024). 2D-ANEURYSM, 3D-ANEURYSM were generated with CimLib Digonnet et al. (2007) and were introduced by Goetz et al. (2024b).

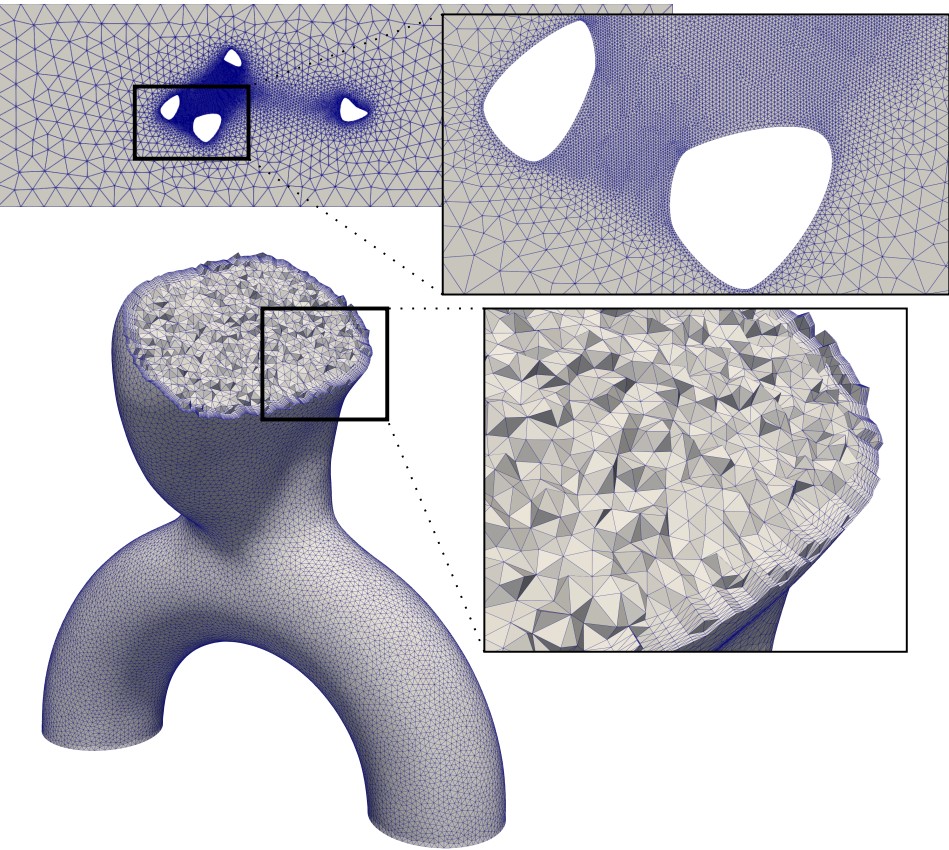

Figure 9: Overview of our unstructured mesh. **(top)** Mesh from the Bezier dataset with around 30k nodes. **(bottom)** Mesh from the 3D-Aneurysm dataset with around 250k nodes.

| Dataset | Inputs | Outputs | History | $t_{gnn}$ **(ms/step)** | $t_{gt}$ **(ms/step)** |
|---|---|---|---|---|---|
| CYLINDER | $n, v_x, v_y$ | $v_x, v_y$ | 0 | 33 | 820 |
| PLATE | $n, x, y, z, f_{\text{in}}$ | $x, y, z, \sigma$ | 0 | 36 | 2893 |
| FLAGSIMPLE | $n, x, y, z, f_{\text{in}}$ | $x, y, z$ | 1 | 22 | 4166 |
| AIRFOIL | $n, v_x, v_y, \rho$ | $v_x, v_y, \rho$ | 0 | 46 | 11015 |
| BEZIER | $n, v_x, v_y$ | $v_x, v_y$ | 0 | 57 | 986 |
| 2D-ANEURYSM | $n, v_x, v_y, v_{\text{in}}$ | $v_x, v_y$ | 1 | 243 | - |
| 3D-ANEURYSM | $n, v_x, v_y, v_z, v_{\text{in}}$ | $v_x, v_y, v_z$ | 1 | 981 | 540000 |

In A.1.1, $n$ is the node type, $f_{\text{in}}$ the force being applied at the current time step to the object and $v_{\text{in}}$ the inflow velocity at the current timestep. When history is different than 0, we use a first-order derivative of the inputs as extra feature. For example, we also add $a_x, a_y, a_z$ to each node from an aneurysm mesh.

### A.1.2   NOISE

Following the same strategy as Sanchez-Gonzalez et al. (2020), we make our inputs noisy. More specifically, we add random noise $\mathcal{N}(0, \sigma)$ to the dynamical inputs. Each noise was either selected from previous papers, or selected by looking at average one-step error in predictions. Noises are presentend in Table A.1.2.

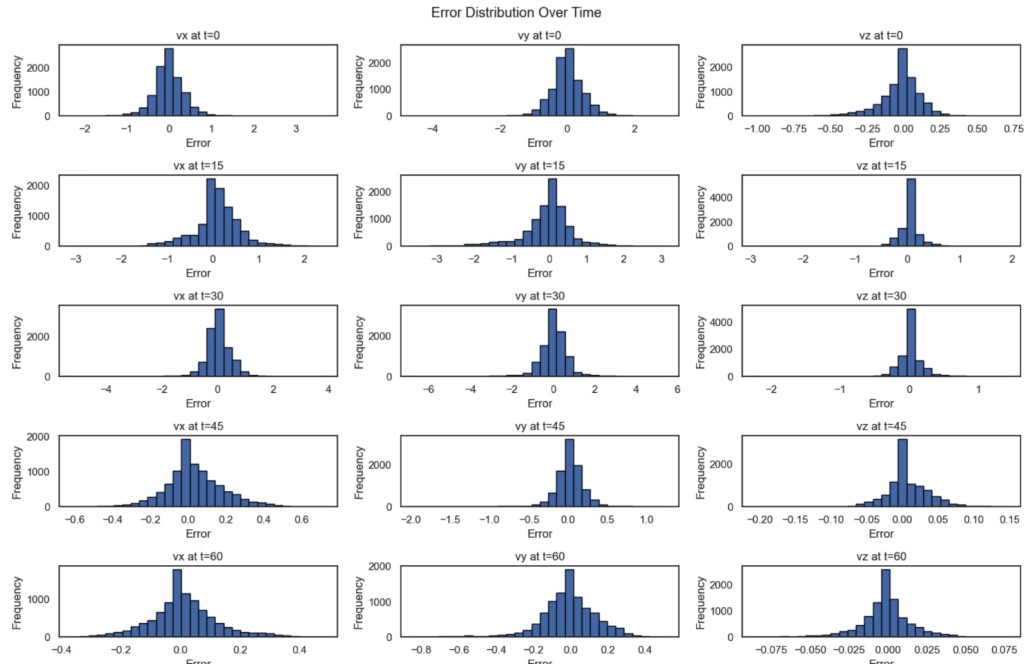

Figure 10: Error distribution on different timesteps in a trajectory predicted from the 3D-Aneurysm dataset. The magnitude evolves both with the axis and with the time.

| Dataset | Noise | Sub-mesh |
|---|---|---|
| CYLINDER | 0.02 | - |
| PLATE | 0.003 | - |
| FLAGSIMPLE | 0.001 | - |
| AIRFOIL | 10 | - |
| BEZIER | 0.02 | - |
| 2D-ANEURYSM | 10 | - |
| 3D-ANEURYSM | $v_x, v_y$: 10, $v_z$: 0.5 | 100000, 10 |

### A.1.3 ERROR DISTRIBUTION

In the case of 2D-ANEURYSM and 3D-ANEURYSM, since the velocity inflow is different depending on the timesteps, the error distribution varies a lot (see Figure 10).

While this did not prove to be an issue in 2D, we had more trouble to find an acceptable noise in 3D. One solution that was tried but proved to be unsuccessful was to use dynamic noise depending on the time steps, using the error distribution seen above. This led to similar results to the static noise we end up using.

### A.2 SUB-MESH PARTITIONING

Some meshes from the 3D-ANEURYSM dataset were too large to fit in GPU memory and thus necessitated to be split into sub-meshes. The strategy applied was to split one mesh into smaller sub-meshes, and instead of one gradient descent apply multiples ones on each of the sub-mesh. Some sub-meshes are presented Figure 11.

Sub-meshes were generated using two strategies: a random neighbor sampling strategy from Hamilton et al. (2018) using between 50000 and 100000 random edges. The second strategy used the METIS Padua et al. (2011) algorithm to generate between 7 and 15 disjoint sub-meshes.

We conducted an extensive study to compare results on model trained with and without this sub-meshs strategy and found no meaningful difference in accuracy.

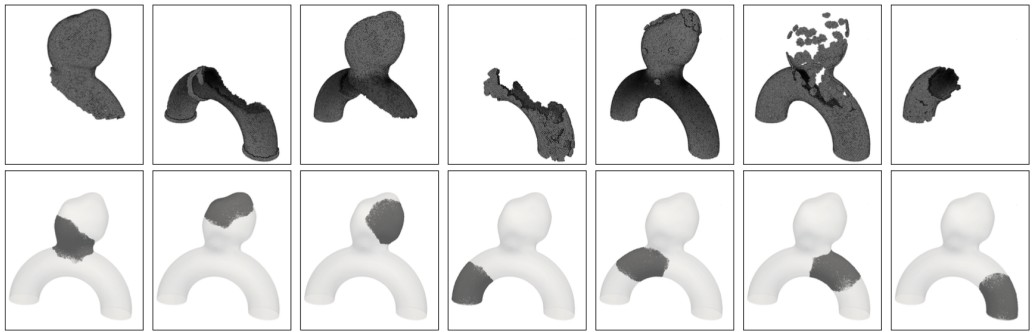

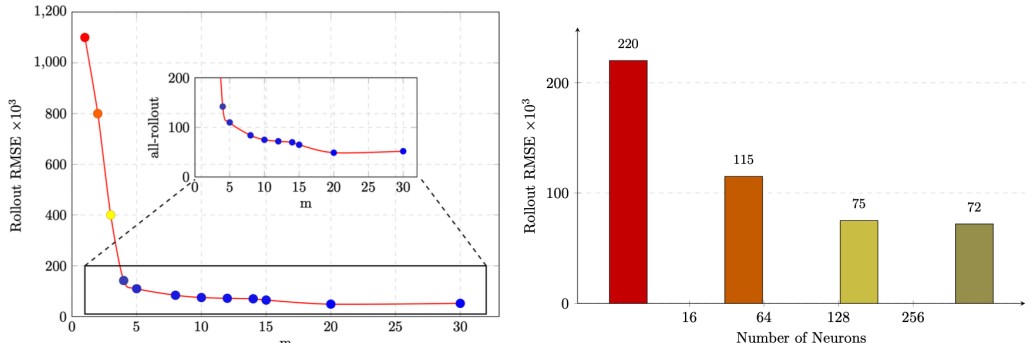

Figure 11: Different sub-mesh generated at each batch update.

Figure 12: **(left)** Ablation study of the number of message passing steps. We encounter a plateau starting from $m = 10$. **(right)** Impact of the number of neurons per layer. We encounter a plateau starting from $n = 128$.

### A.3 ABLATION STUDY

We also conducted an ablation study on the Cylinder dataset regarding the number of message passing steps and the number of neurons per layer. Results can be seen in Figure 12.

### A.4 ADDITIONAL RESULTS

Additional Results are presentend in Figures 13 and 14.

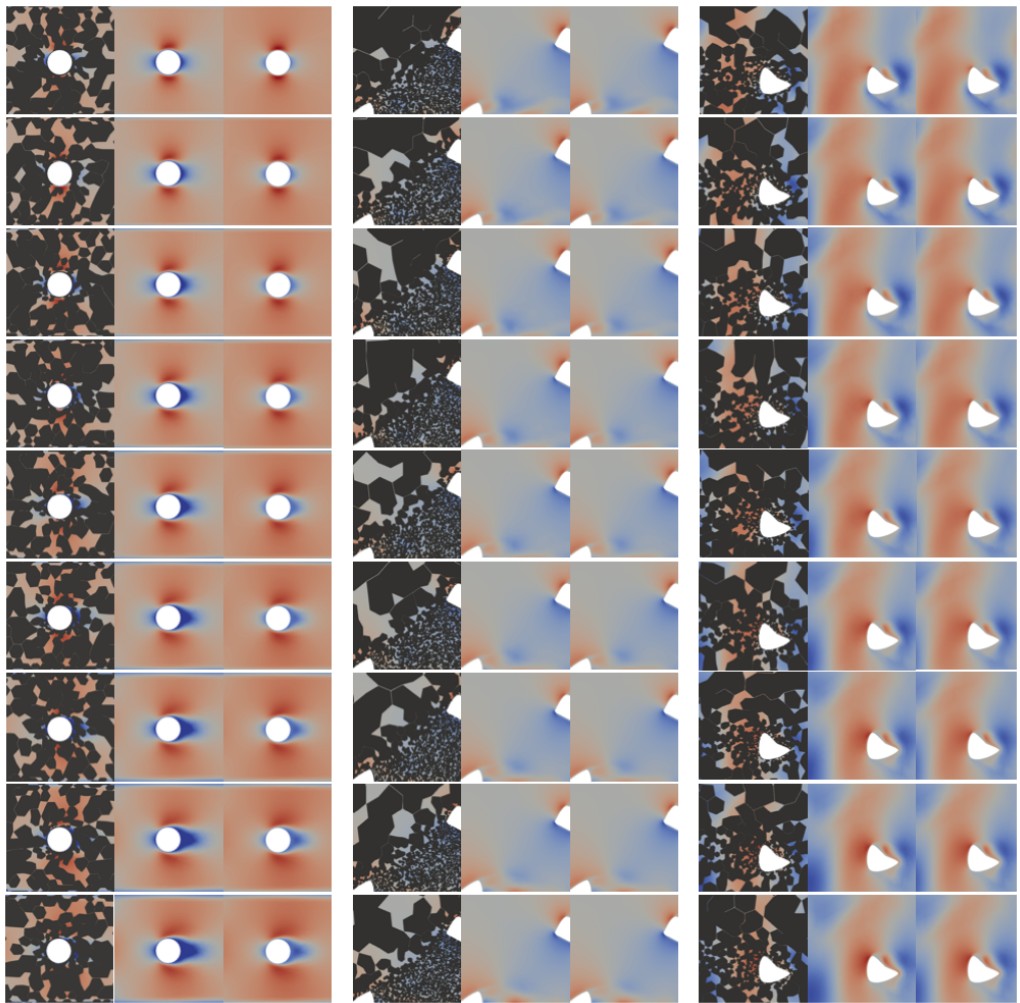

Figure 13: Uncurated random shapes from the validation cylinder and bezier shapes. **(left)** masked, **(middle)** predicted and **(right)** original.

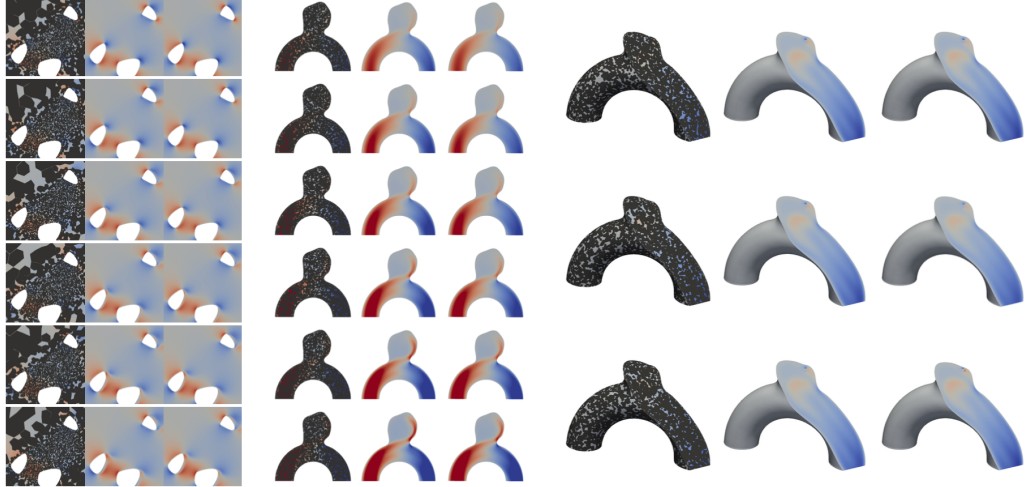

Figure 14: Uncurated random shapes from the validation aneurysm and bezier shapes. **(left)** masked, **(middle)** predicted and **(right)** original.

