# OpenReview forum: "MeshMask: Physics-Based Simulations with Masked Graph Neural Networks"
_ICLR.cc/2025/Conference — ICLR 2025 Poster_

### Official Review · Reviewer_gGsd · 2024-10-28

**Soundness:** 3
**Presentation:** 3
**Contribution:** 3
**Rating:** 8
**Confidence:** 3

**Summary:**

This paper introduces MeshMask, a GNN masking pre-training technique designed to enhance the efficiency and performance of GNNs in solving CFD problems. Drawing inspiration from the success of masked autoencoders in NLP and CV, the authors adapt masking pre-training techniques to GNN models in the CFD domain. MeshMask demonstrates performance improvements across multiple CFD datasets.

**Strengths:**

1. The authors developed an appropriate masking pre-training scheme for GNNs in CFD tasks.

2. The method shows notable performance improvements compared to baseline approaches.

3. The paper is well-organized, with clear figures and visualizations that help the reader understand.

**Weaknesses:**

1. The comparison seems limited to very basic baseline methods~(i.e., MGN, Multigrid). Given the recent attention to GNN-based CFD modeling, more recent baselines could have been included in the comparison.

2. Despite multiple claims about *training efficiency*, there appears to be a lack of comprehensive discussion on efficiency metrics.

3. The experiments lack testing across varying *graph density* scenarios. The effectiveness of the method across different graph densities could have been better demonstrated.

**Questions:**

1. Please discuss and address the concerns raised in the weaknesses.

2.  Why is the transfer learning on CFDs effective? Could the authors provide some high-level discussion about the underlying mechanisms?

---

> ### Author Response · Authors · 2024-11-18
>
> Thank you for your comments, questions and suggestions to improve the manuscript.
>
> - > The comparison seems limited to very basic baseline methods~(i.e., MGN, Multigrid). Given the recent attention to GNN-based CFD modeling, more recent baselines could have been included in the comparison.
>
> We fully agree with the reviewer on that matter. We realized this soon after the draft submission and immediately started training other architectures on the same task. We are currently finishing the training and will add their performances to our main table. Overall, even compared to more recent approaches, our methods still yield significant improvements. It is also important to note that our pre-training technique is fully architecture agnostic and can be used as a plug-and-play technique with any GNN architecture.
>
> - > Despite multiple claims about training efficiency, there appears to be a lack of comprehensive discussion on efficiency metrics.
>
> We agree that the paper lacks information regarding proof of training efficiency in its current state. We are adding details regarding the number of parameters of each model, the amount of vRAM used, and training and inference time. We are also adding details regarding training time and comparing it with computation time with classical Finite Element and Finite Volume methods.
> Mainly, our findings are the following:
>
> - Even when taking into account the time needed to build the dataset as well as the training time, it takes around 100 simulations of aneurysms for our machine learning method to become more efficient
> - Even with more parameters due to the GatedMLP, our architecture retains similar training and inference as MeshGraphNet [1]
> - Even during the pretraining phase where our overall model is much larger than previous methods (3 more message passing steps), due to the fact that the graphs we work with are much smaller (around 65\% smaller) thanks to masking nodes and edges, our models retains similar training and inference as MeshGraphNet [1]
>
> - > The experiments lack testing across varying graph density scenarios. The effectiveness of the method across different graph densities could have been better demonstrated.
>
> We fully agree with the reviewer regarding the lack of testing on graphs of varying density. This is mostly due to the way meshes are built when dealing with classical Finite Element and Finite Volume methods. When building a mesh (and thus a graph) for such application, the goal is to optimize the cell qualities of the graph (such as the Frobenius Ratio or the Aspect ratio). Even when varying the mesh refinement (i.e. the maximal edge length of the graph), one ends up with meshes of similar density and average degree.
> However, we ran several experiments in which we trained a model on graphs with a specific refinement (say 10,000 nodes, for example) and then applied this model directly to a much finer graph (say 250,000 nodes, for example). We find that even with this large gap in refinement, our model is able to generalize well and make accurate predictions. This also means that we do not necessarily have to produce the entirety of the training on large and costly meshes, but that a fraction is sufficient.
>
> [1] : Learning Mesh-Based Simulation with Graph Neural Networks

---

> > ### Author Response · Authors · 2024-11-18
> > **(continue)**
> >
> > - > Why is the transfer learning on CFDs effective? Could the authors provide some high-level discussion about the underlying mechanisms?
> >
> > While our objective was not to try to train a model capable of predicting flow fields for all our datasets because of their fundamental differences in flow characteristics and underlying physics, we managed to yield good results with the CYLINDER and BEZIER datasets for several reasons.
> > The first one would be the similarity between both datasets;  the BEZIER dataset is only different from the CYLINDER one through the fact that the flow is not going around a cylinder, but several randomly generated rigid bodies. One could argue that the mechanisms learned by our network are close enough in both datasets so that transfer learning can apply.
> > The second and maybe most important one would be that both datasets are created through the resolution of the physical problem, which is to solve the incompressible Navier Stokes equations for a Newtonian fluid, with a no-slip condition at the wall and obstacle boundaries, steady inflow velocity and within a given range of Reynolds number.
> > \begin{eqnarray}
> >     \rho (\partial_t v +  v \cdot \nabla v) = -\nabla p + \mu \Delta v \\
> > \end{eqnarray}
> > \begin{eqnarray}
> >     \nabla \cdot v = 0
> > \end{eqnarray}
> > Much like FEM solvers working locally in the computational domain, we could argue that because of their shared physics, both datasets are highly similar when looking at local areas where message-passing mechanisms operate, thus leading to a model learning to solve the Navier Stokes equations in this given physical setting.

---

> ### Comment · Reviewer_gGsd · 2024-11-25
>
> I thank the authors for their well-organized response and revision. I am pleased to see the paper's improvements and have upgraded my evaluation scores. Good luck!

---

> > ### Author Response · Authors · 2024-11-26
> >
> > Dear Reviewer gGsd,
> >
> > We deeply appreciate your support and the valuable feedback you provided to enhance our paper.
> > Thank you once again!
> >
> > Best regards,
> >
> > The Authors

---

### Official Review · Reviewer_i9XS · 2024-10-30

**Soundness:** 3
**Presentation:** 2
**Contribution:** 2
**Rating:** 5
**Confidence:** 4

**Summary:**

The paper introduces a masked pre-training technique for Graph Neural Networks (GNNs) aimed at solving physics simulations, particularly in computational fluid dynamics. The proposed approach uses an asymmetric encoder-decoder architecture in conjunction with gated multi-layer perceptrons. Extensive experiments are conducted, including independent training on multiple datasets, transfer learning, and multi-dataset pretraining. The results demonstrate that the MeshMask model delivers competitive performance compared to established baselines.

**Strengths:**

The paper introduces a novel approach by implementing masked pre-training for Graph Neural Networks (GNNs) in physics simulations. It is well-structured and easy to follow, with rich qualitative results and visualizations.

**Weaknesses:**

1. The introduction of gated MLP and encoder-decoder architecture potentially increases the model's computational complexity. There is a lack of detailed discussion on the computational demands (e.g. training/inference time,  training/inference RAMs) compared with baselines, which is critical for evaluating the feasibility of deploying MeshMask in real-time applications or on large-scale datasets.
2. The compared baselines are limited, authors should consider more GNN-based simulation models such as [1-4]. Furthermore, both GCN and U-Net baselines have only been tested on a few datasets.
3. The novelty of replacing standard MLPs with gated MLPs in the processor is somewhat limited. The authors should consider a wider variety of network architectures such as attention-based models [1,2,4] for comparison to make their findings more persuasive.

[1] Eagle: Large-scale learning of turbulent fluid dynamics with mesh transformers.

[2] Learning flexible body collision dynamics with hierarchical contact mesh transformer.

[3] Efficient Learning of Mesh-Based Physical Simulation with BSMS-GNN.

[4] Transformer with Implicit Edges for Particle-based Physics Simulation.

**Questions:**

1. Given that your input mesh includes masked data, how do you effectively learn long-range interactions on the masked graph during the repeated message passing?
2. In Sec 4.3,  given that the input and output physics quantities differ across the datasets, the potential for a significant gap in the encoded latent space raises concerns about the legality and effectiveness of the transfer learning approach. How do you ensure that the latent spaces of the two datasets are compatible? Secondly, the length of input and output quantities varies across datasets; how do you exactly fine-tune the model when faced with these differing lengths?
3. What is the model's performance on out-of-distribution (OOD) mesh resolutions?

---

> ### Author Response · Authors · 2024-11-18
>
> Thank you for your comments, questions and suggestions to improve the manuscript.
>
> - > The introduction of gated MLP and encoder-decoder architecture potentially increases the model's computational complexity. There is a lack of detailed discussion on the computational demands (e.g. training/inference time, training/inference RAMs) compared with baselines, which is critical for evaluating the feasibility of deploying MeshMask in real-time applications or on large-scale datasets.
>
> We agree that the paper lacks information regarding computational complexity in its current state. We are adding details regarding the number of parameters of each model, the amount of vRAM used, and training and inference time. We are also comparing training and inference times with classical Finite Element and Finite Volume methods. Finally, we are also adding those parameters in the ablation study when comparing performances between classical MLP and GatedMLP.
> Mainly, our findings are the following:
>
> - Even when taking into account the time needed to build the dataset as well as the training time, it takes around 100 simulations of aneurysms for our machine learning method to become more efficient
> - Even with more parameters due to the GatedMLP, our architecture retains similar training and inference as MeshGraphNet [5]
> - Even during the pretraining phase where our overall model is much larger than previous methods (3 more message passing steps), due to the fact that the graphs we work with are much smaller (around 65\% smaller) thanks to masking nodes and edges, our models retains similar training and inference as MeshGraphNet [5]
>
> - > The compared baselines are limited, authors should consider more GNN-based simulation models such as [1-4]. Furthermore, both GCN and U-Net baselines have only been tested on a few datasets.
>
> We fully agree with the reviewer on that matter. We realized this soon after the draft submission and immediately started training other architectures on the same task. We are currently finishing the training and will add their performances to our main table. We also agree that the comparison with GCN and U-Net are somehow limited. They were added as historical comparisons but since they are now largely outdated in comparison to many different architectures, we plan on removing them from our main table.
> Overall, our methods still yield significant improvements compared to more recent approaches. It is also important to note that our pre-training technique is fully architecture agnostic and can be used as a plug-and-play technique with any GNN architecture.
>
> - > The novelty of replacing standard MLPs with gated MLPs in the processor is somewhat limited. The authors should consider a wider variety of network architectures such as attention-based models [1,2,4] for comparison to make their findings more persuasive.
>
> We fully agree that simply comparing standard MLPs and gatedMLP is not enough to prove the usefulness of the approach. We followed the reviewer’s advice and replaced our gatedMLP with an attention-based mechanism (mostly based on GAT [6]). We are adding a comparison with such an approach (i.e. replacing the gatedMLP by a GAT layer) in the ablation study. Overall, we find such attention-based approach to yield similar results to the gatedMLP (thus having both outperforming the standard MLPs, for a similar training time and computational complexity).
> We also agree with the reviewer with the fact that simply replacing the standard MLP by another building block is a limited improvement. We also consider it to be a minor improvement in comparison to the masking pre-training technique. We can see how the addition of a gated MLP can make the paper a little bit noisy and distract readers from the main results. We are inclined to follow the reviewer’s advice on that matter and can make the parts related to the gated MLP minimal if needed.
>
> [5] : Learning Mesh-Based Simulation with Graph Neural Networks
>
> [6] : Graph Attention Networks

---

> > ### Author Response · Authors · 2024-11-18
> > **(continue)**
> >
> > - > Given that your input mesh includes masked data, how do you effectively learn long-range interactions on the masked graph during the repeated message passing?
> >
> > We agree with the reviewer that this is a very important aspect that we did not discuss in the submitted draft. To ensure that even with removed edges, the model could properly process long-range interactions with message-passing, we consider 2 different approaches:
> >
> > - Not doing anything, and assuming the model is strong enough to learn even with disconnected sub-graphs.
> > - After removing edges, compute the $K$-hop of each remaining node in the original graphs and add those new longer edges if both nodes were not masked. This approach was tested with $K=2$ and $K=3$.
> >
> > We find that adding $K$-hop edges did not improve performance for small masking values (up to 20%). However, for larger values, adding edges with $K=2$ and then $K=3$ performs better. We will update the paper with those details.
> >
> > - > In Sec 4.3, given that the input and output physics quantities differ across the datasets, the potential for a significant gap in the encoded latent space raises concerns about the legality and effectiveness of the transfer learning approach. How do you ensure that the latent spaces of the two datasets are compatible? Secondly, the length of input and output quantities varies across datasets; how do you exactly fine-tune the model when faced with these differing lengths?
> >
> > We thank the reviewer for this remark  and fully agree that we could add explanations on our transfer learning experiment.
> > While we present datasets in this paper that do not necessarily share input and output quantities, or that can even differ from a physics standpoint, we ran our transfer learning experiments on 2 similar datasets. The CYLINDER dataset contains two-dimensional fluid mechanics simulations of a flow in a rectangular domain, going past a cylinder; the BEZIER dataset is only different from the CYLINDER one through the fact that the flow is not going around a cylinder anymore, but a few random shapes generated from Bézier curves. The BEZIER dataset also contains longer trajectories; the simulations ran for a higher number of timesteps to make sure that the flow would stabilize even in cases with more complex geometries used as obstacles.
> > Both datasets are created by solving the incompressible Navier Stokes equations for a Newtonian fluid, with a no-slip condition at the wall and obstacle boundaries, with a steady inflow and within the same range of Reynolds number. They both use the two-dimensional velocity field and the node type as inputs and return the next-step two-dimensional velocity field as output.
> > \begin{eqnarray}
> >     \rho (\partial_t v +  v \cdot \nabla v) = -\nabla p + \mu \Delta v \\
> > \end{eqnarray}
> > \begin{eqnarray}
> > \nabla \cdot v = 0
> > \end{eqnarray}
> > Because the flow simulations in both datasets share the same characteristics, governing equations, settings, similar boundary conditions, and most importantly, the same inputs and output dimensions, we can confidently ensure that a transfer learning approach is relevant here and that the latent spaces built from training on each of these datasets are similar enough to justify such an approach.
> >
> > - > What is the model's performance on out-of-distribution (OOD) mesh resolutions?
> >
> > We agree with the reviewer that this is an interesting axis to consider. We ran several experiments in which we trained a model on graphs with a specific refinement (say 10,000 nodes, for example) and then applied this model directly to a much finer graph (say 250,000 nodes, for example). We find that even with this large gap in refinement, our model is able to generalize well and make accurate predictions (a model trained on a coarse mesh and evaluated on fine meshes performs 75% worse than a model trained fully on fine mesh, which remains on par with results from MeshGraphNet for example). This also means that we do not necessarily have to produce the entirety of the training on large and costly meshes, but that a fraction is sufficient.

---

> > > ### Comment · Reviewer_i9XS · 2024-11-25
> > >
> > > Thank you for your detailed response. I've read the responses and most of my concerns have been addressed. I have raised my score (3 --> 5).

---

> > > > ### Author Response · Authors · 2024-11-26
> > > >
> > > > Dear Reviewer i9XS,
> > > > Thank you so much for your thoughtful support and for helping us strengthen our paper. Your detailed feedback has been incredibly helpful in refining the presentation and ensuring its quality, especially regarding other baselines and the gated MLP.
> > > > We truly appreciate it!
> > > > Warm regards,
> > > > The Authors

---

### Official Review · Reviewer_thnV · 2024-11-02

**Soundness:** 3
**Presentation:** 3
**Contribution:** 3
**Rating:** 6
**Confidence:** 4

**Summary:**

The paper proposes a masked pretraining strategy for learning mesh-based simulation. An asymmetric encoder/decoder architecture is proposed for doing masked-node reconstruction. Throughout numerical experiment, the authors showcase the pretraining can improve temporal prediction accuracy over other baselines.

**Strengths:**

* The proposed pretraining consistenly improves GNN’s performance on various physical systems.

* The numerical experiments are extensive and cover relatively challenging problems. Specifically, a large-scale 3D simulation with 250k nodes is studied.

**Weaknesses:**

* The discussion related to the model architecture and the graph construction after masking is relatively vague. The authors say they used a multi-grid GNN with a W-cycle structure and is a Encoder-Processor-Decoder setup, but it is also stated that in the pretraining section the model for pertaining is an Autoencoder with graph encoder/decoder. Does the model architecture and grid structure change or remain the same in pretraining and fine-tuning? In addition, as the masking nodes are randomly sampled, is the masked mesh still valid after an excessively high masking ratio?

* For the numerical experiments, the authors describes the detail for pre-training and fine-tuning of their proposed model, but it seems that there is not too much information in terms of the training budget of the baseline model (or I have missed them). For example, suppose training a GNN on next-step prediction without masking for 100k steps needs 1e9 flops, and then pretraining it for 100k steps needs another 1e9 flops. Then for the baseline models without pretraining, a fair comparison would be training them using 2e9 flops.

* Following the last point, if the pretraining can be done on a dataset order-of-magnitude larger than almost all downstream tasks (for example, LLMs), then maintaining similar flops between pretraining/no pretraining model indeed becomes meaningless. However, according to the result shown in Table 3, pretraining on a mutli-physics dataset actually performs worse than pretraining on the same physics, and pretraining on a completely different physics is actually harmful to the downstream tasks.

**Questions:**

* Does the masking algorithm consider irregular connectivity? For example, a sub-graph is detached from other sub-graphs after masking.

* Following point 2 in the Weakness, can authors provide more details in terms of the training time/ number of parameters of different models?

---

> ### Author Response · Authors · 2024-11-18
>
> Thank you for your comments, questions and suggestions to improve the manuscript.
>
> - > The discussion related to the model architecture and the graph construction after masking is relatively vague. The authors say they used a multi-grid GNN with a W-cycle structure and is a Encoder-Processor-Decoder setup, but it is also stated that in the pretraining section the model for pertaining is an Autoencoder with graph encoder/decoder. Does the model architecture and grid structure change or remain the same in pretraining and fine-tuning? In addition, as the masking nodes are randomly sampled, is the masked mesh still valid after an excessively high masking ratio?
>
> We thank the reviewer for this remark, we revised the paper accordingly by including details regarding the full architecture and more description on the masking strategy
> Our model model is composed of 2 graph neural networks; an Encoder and a Decoder, that we could call “global Encoder and Decoder” here for the sake of clarity. Each of these 2 models is in fact a multi-grid graph neural network built in a Encode-Process-Decode fashion. During the masked pretraining, we train both the global Encoder and Decoder, and we then fine-tune only the global Encoder; the model architecture does not change between pretraining and fine-tuning. We will add details and clarifications to our paper on this point.
> Regarding the point about masking, it is true that after a large masking ratio is applied, the output graph could not be used as a mesh per-se for a classical Finite Volume method. Similarly, having such disconnected nodes brings the question of the relevance of message passing and if the information can properly flow through different nodes. We detail how we tackled this in the answer to your first question !
>
> - > For the numerical experiments, the authors describes the detail for pre-training and fine-tuning of their proposed model, but it seems that there is not too much information in terms of the training budget of the baseline model (or I have missed them). For example, suppose training a GNN on next-step prediction without masking for 100k steps needs 1e9 flops, and then pretraining it for 100k steps needs another 1e9 flops. Then for the baseline models without pretraining, a fair comparison would be training them using 2e9 flops.
>
> That is a perfectly valid point that misses clarity in the draft, and we thank the reviewer for pointing it out. To have a valid comparison between all our models, pretrained or not, we made sure to train every model for a total of one million training steps. The models pretrained with masking were pretrained for 500k training steps, and then trained for an additional 500k steps.
> The revised version now makes all these details available, particularly details about the training cost and the amounts of VRAM used.
>
> - > Following the last point, if the pretraining can be done on a dataset order-of-magnitude larger than almost all downstream tasks (for example, LLMs), then maintaining similar flops between pretraining/no pretraining model indeed becomes meaningless. However, according to the result shown in Table 3, pretraining on a mutli-physics dataset actually performs worse than pretraining on the same physics, and pretraining on a completely different physics is actually harmful to the downstream tasks.
>
> Indeed, we have revised the paper accordingly by incorporating details about the transfer learning. As mentioned in the remark, we are not in the first mentioned case of a model pretrained on a dataset much larger than downstream tasks, but rather on two datasets at most, including the one concerned by the downstream task. It is also worth noting that all the compared models are trained for a total of a million training steps and that our baseline comparison for both experiments is a model simply trained on a single dataset.
>
> While pretraining a model on a single physics dataset different from the one in the downstream task is harmful to its performance, we show that pretraining on a multi-physics dataset and fine-tuning on a single dataset yield better results than our baseline.
> Taking the example of next-step prediction on the BEZIER dataset, pretraining on both the BEZIER and CYLINDER dataset for 500k training steps and fine-tuning on the BEZIER dataset for another 500k training steps produce a better model than training for 1 million training steps on the BEZIER dataset. Quite logically, these results remain lower than those of a model pretrained only on BEZIER and fine-tuned on BEZIER as well, but we show that our pre-training method allows the pretraining of a model on multiple datasets before being fine-tuned and still generates better performances than simple training.
>
> This allows for better performance on 2 datasets in only 1.5m steps compared to 2m steps on the said 2 datasets, thus saving 33% of the time while improving the performances. All these details are now added to the revised version.

---

> > ### Author Response · Authors · 2024-11-18
> > **(continue)**
> >
> > - > Does the masking algorithm consider irregular connectivity? For example, a sub-graph is detached from other sub-graphs after masking.
> >
> > We agree with the reviewer that this is a very important point that requires more attention in the new revised version. We considered 2 different approaches:
> >
> > - Not doing anything, and assuming the model is strong enough to learn even with disconnected sub-graphs.
> > - After removing edges, compute the $K$-hop of each remaining node in the original graphs and add those new longer edges if both nodes were not masked. This approach was tested with $K=2$ and $K=3$.
> >
> > We find that adding $K$-hop edges did not improve performance for small masking values (up to 20\%). However, for larger values, adding edges with $K=2$ and then $K=3$ performs better. We will update the paper with those details.
> >
> > - > Following point 2 in the Weakness, can authors provide more details in terms of the training time/ number of parameters of different models?
> >
> > We agree that the paper lacks information regarding computational complexity in its current state. We have added details regarding the number of parameters of each model, the amount of vRAM used, and training and inference time. We also added details regarding training time and compared it with computation time with classical Finite Element and Finite Volume methods.
> > Mainly, our findings are the following:
> >
> > - Even when taking into account the time needed to build the dataset as well as the training time, it takes around 100 simulations of aneurysms for our machine learning method to become more efficient
> > - Even with more parameters due to the GatedMLP, our architecture retains similar training and inference as MeshGraphNet [1]
> > - Even during the pretraining phase where our overall model is much larger than previous methods (3 more message passing steps), due to the fact that the graphs we work with are much smaller (around 65\% smaller) thanks to masking nodes and edges, our models retains similar training and inference as MeshGraphNet [1]
> >
> > [1] : Learning Mesh-Based Simulation with Graph Neural Networks

---

> > > ### Comment · Reviewer_thnV · 2024-11-25
> > > **Reply to authors**
> > >
> > > I would like to thank the authors for their efforts. I've read the responses and am glad to see the update to the manuscript, which has clarified and addressed most of my concerns. I think this work will be interesting for practitioners in GNN + mesh-based simulation community, so I have adjusted my score accordingly.

---

> > > > ### Author Response · Authors · 2024-11-26
> > > >
> > > > Dear Reviewer thnV,
> > > >
> > > > We are truly grateful for your assistance and constructive feedback. Your insights have been very helpful in enhancing the overall quality of our paper and improving its presentation.
> > > >
> > > > Best wishes,
> > > >
> > > > The Authors

---

### Author Response · Authors · 2024-11-19
**Revised version**

We thank the reviewers for their questions and feedback to improve the manuscript. We have revised our paper and uploaded a new version. All new and updated texts are marked in Magenta in the pdf.
We added suggested edits and clarifications and also ran new experiments regarding other baselines and out-of-distribution meshes, as requested and recommended by the reviewers.

More precisely, the modifications are the following:
- trained additional baselines (BSMS-GNN [1] and MultiScale-GNN [2]) on the same datasets. Overall, they perform similarly to the Multigrid architecture, and our Masking pretraining technique still yields large improvement
- reported quantitative numbers regarding training efficiency. More specifically, we added a new table reporting the number of training steps, parameters, the vRAM usage, and the inference time
- clarified the transfer learning experiments to highlight the similarity between the considered datasets and the added value of our pretraining method
- trained a model on coarse meshes (10k nodes) to evaluate it on very fine meshes (250k nodes) to quantify how well our architecture and methods perform on out-of-distribution meshes
- added more details regarding the management of disconnected sub-graphs in the case of a large masking ratio
- ran a model with Graph Attention Layers instead of GatedMLP. It performs very similarly to the GatedMLP version, making them both exchangeable
- ran an MGN architecture with our pretraining method. We find that this offers up to 25% improvements over the classical MGN architecture, and bridges the gap with more recent methods such as [1] and [2]
- clarified the overall architecture, and potential confusion between AutoEncoder and Encode-Process-Decode

We believe this revision should address all points raised in the review. Videos were updated at the following url: https://sites.google.com/view/masked-graph-neural-networ/

[1] : Efficient Learning of Mesh-Based Physical Simulation with BSMS-GNN

[2] : Simulating Continuum Mechanics with Multi-Scale Graph Neural Networks

---

> ### Author Response · Authors · 2024-11-24
>
> We really believe that our pre-training method can improve the results of many different GNN architecture. We hope our updates truly answers every reviewers concerns and questions.
> We updated the paper and all new and modified texts are marked in Magenta in the pdf.
> Please let us know if you still have concerns about some parts of the methodology.

---

### Meta-Review · Area_Chair_rx7C · 2024-12-19

**Metareview:**

The paper introduces an interesting technique of randomly masking the graph nodes in graph neural networks, specifically in learned simulation for fluid dynamics. The method consistently improves prediction accuracy on challenging problems, including large-scale 3D simulation with 250k nodes. The method shows notable performance improvements compared to baseline approaches. The paper also includes experiments on transfer learning between datasets and multi-dataset pretraining. Although reviewers had questions about training efficiency in the original paper, the authors have effectively addressed them in the rebuttal. I recommend the paper for Accept (spotlight).

**Additional Comments On Reviewer Discussion:**

To respond to reviewers' concerns, the authors added several new baselines and reported quantitative numbers on training efficiency.

---

### Decision · Program_Chairs · 2025-01-22

Accept (Poster)